# Association between metabolically healthy obesity/overweight and cardiovascular disease risk: A representative cohort study in Taiwan

Tzu-Lin Yeh[1,2], Hsin-Yin Hsu[2,3], Ming-Chieh Tsai[2,4,5], Le-Yin Hsu[2], Lee-Ching Hwang[3,5], Kuo-Liong Chien[2,6]*

1 Department of Family Medicine, Hsinchu MacKay Memorial Hospital, Hsinchu, Taiwan, 2 Institute of Epidemiology and Preventive Medicine, National Taiwan University, Taipei, Taiwan, 3 Department of Family Medicine, Taipei MacKay Memorial Hospital, Taipei, Taiwan, 4 Division of Endocrinology, Department of Internal Medicine, MacKay Memorial Hospital, Tamsui Branch, New Taipei City, Taiwan, 5 Department of Medical Research, MacKay Memorial Hospital, New Taipei City, Taiwan, 6 Department of Internal Medicine, National Taiwan University Hospital, Taipei, Taiwan

* klchien@ntu.edu.tw

**Data Availability Statement:** The data underlying the current study are not publicly available due to the terms of consent to which the participants agreed. Data are available from the National Taiwan

## Abstract

### Objectives

To investigate the relationship between metabolically healthy obesity and cardiovascular disease risk in Taiwanese individuals.

### Methods

Taiwanese individuals were recruited from a nationwide, representative community-based prospective cohort study and classified according to body mass index as follows: normal weight (18.5–23.9 kilogram (kg)/meter(m)$^2$) and obesity/overweight ($\geq$24 kg/m$^2$). Participants without diabetes, hypertension, and hyperlipidemia and who did not meet the metabolic syndrome without waist circumference criteria were considered metabolically healthy. The study end points were cardiovascular disease morbidity and mortality. Multivariable adjusted hazard ratios and 95% confidence intervals were obtained from a Cox regression analysis.

### Results

Among 5 358 subjects (mean [standard deviation] age, 44.5 [15.3] years; women, 48.2%), 1 479 were metabolically healthy with normal weight and 491 were metabolically healthy with obesity. The prevalence of metabolically healthy obesity/overweight was 8.6% in the Taiwanese general population, which included individuals who were >20 years old, not pregnant, and did not have CVD (n = 5,719). In the median follow-up period of 13.7 years, 439 cardiovascular disease events occurred overall and 24 in the metabolically healthy obesity group. Compared with the reference group, the metabolically healthy obesity group had a

University Hospital Institutional Data Access /
Ethics Committee (contact via ntuepm@ntu.edu.
tw) for researchers who meet the criteria for
access to confidential data or from the authors
upon reasonable request and with permission of
the Health Promotion Administration at the
Ministry of Health and Welfare in Taiwan. (https://
dep.mohw.gov.tw/DOS/cp-2516-3591-113.html;
https://dep.mohw.gov.tw/DOS/cp-2499-45896-
113.html).

**Funding:** The authors received no specific funding
for this work.

**Competing interests:** The authors have declared
that no competing interests exist.

significantly higher cardiovascular disease risk (adjusted hazard ratio: 1.74, 95% confidence interval: 1.02, 2.99).

## Conclusions

Individuals with metabolically healthy obesity have a higher risk of cardiovascular disease and require aggressive body weight control for cardiovascular disease control.

## Introduction

Cardiovascular disease (CVD) is the leading cause of mortality worldwide. According to the World Health Organization, over million deaths occurred due to CVD in 2017 [1] and the number is increasing. Obesity and overweight are well-known risk factors for CVD due to metabolic dysregulation. However, some obesity phenotypes that are protected from adverse metabolic effects of excess body fat are considered "metabolically healthy" [2]. The concept of metabolically healthy obesity (MHO), indicating that some individuals with obesity do not have any negative health outcomes, was first reported in 2001 [3]. However, whether MHO is a fact or threat, a friend or a foe [4, 5], is being debated. Obesity/overweight status is usually evaluated by body mass index (BMI), waist circumference (WC), or body fat. Individuals with no diagnosis of metabolic syndrome or those who are insulin sensitive or have normal blood pressure, glucose level, and lipid profile are considered metabolically healthy [6, 7]. The prevalence of MHO ranges from 6% to 60% [8, 9], and the relationship between MHO and CVD is uncertain due to the lack of consensus regarding MHO definition [10]. According to the updated definition of metabolic health that was proposed in 2019, metabolically healthy means absence of cardiometabolic diseases and a healthy cardiometabolic blood profile [11].

Recently, a meta-analysis revealed that Asian individuals with MHO had a significantly higher risk of CVD (up to 61%) than individuals with metabolically healthy normal weight (MHNW) [12]. However, the meta-analysis did not include articles from Taiwan. The only one study that reported MHO and its impact on hypertension, diabetes, and metabolic syndrome in Taiwanese population [13] lacked CVD outcomes. Furthermore, potential effect modifiers may exist according to previous studies [14]. Thus, we investigated the relationship between MHO and CVD risk in Taiwanese individuals and explored potential effect modifiers.

## Materials and methods

### Study design and study participants

We recruited individuals from the Taiwanese Survey on Hypertension, Hyperglycemia, and Hyperlipidemia (TwSHHH), a nationwide, representative community-based prospective cohort study. The BMI category and metabolic status of 6,706 participants were determined based on their basic sociodemographic characteristics and vital sign and biochemical measurements in the first survey in 2002. The second survey was conducted five years later, and its details have been published previously [15]. Participants of the first survey who were more than 20 years old were included; however, those with missing BMI data and who were pregnant, underweight, or had CVD prior to the study were excluded. This study was conducted in accordance with the Declaration of Helsinki and was approved under exempt review

procedures from the Institutional Review Board of the National Taiwan University Hospital (201901103W). Written informed consent was obtained from all the participants.

## Definitions of MHOO

Participants were classified into two BMI categories—normal weight (BMI: 18.5–23.9 kilogram (kg)/meter(m)$^2$) and obesity/overweight (BMI ≥24 kg/m$^2$)—according to the Bureau of Health Promotion, Department of Health, Taiwan [16]. We modified metabolic health definition according to MHO definition published in 2019 [11]. The participants had healthy cardiometabolic profiles and were free from hypertension, type 2 diabetes, and hyperlipidemia. Hypertension was defined as systolic blood pressure ≥140 mmHg and diastolic blood pressure ≥90 mmHg in the first survey or long-term prescription of antihypertensive agents. Type 2 diabetes referred to a fasting glucose level of ≥126 milligram (mg)/ deciliter (dL) and hemoglobin A1c ≥6.5% in the first survey or long-term prescription of antidiabetic agents. Hyperlipidemia referred to a low-density lipid cholesterol (LDL-C) level of ≥160 mg/dL in the first survey or long-term prescription of lipid-lowering agents. The aforementioned prescriptions were retrieved from the National Health Insurance Research Database, which covers 99.9% of the Taiwanese population [17]. Detailed definitions are presented in S1 and S2 Tables. A healthy cardiometabolic profile included all of the following: fasting triglyceride level, <150 mg/dL; high-density lipoprotein cholesterol level, ≥40 mg/dL in men or ≥50 mg/dL in women; fasting glucose level, <100 mg/dL; systolic blood pressure, <130 mmHg; and diastolic blood pressure, <85 mmHg. We did not adopt the criteria of WC since the measurement and the unit used were not consistent in the database. Fasting plasma samples were used for performing all biochemical tests using an automatic analyzer (TBA-200FR, Toshiba Corporation, Tokyo, Japan). Coefficients of variation of these measurements were approximately 5%.

## Outcomes

The study endpoints were fatal and nonfatal CVD (i.e., occurrence of coronary heart events and consequently death) and cerebrovascular morbidity and mortality. Coronary heart events included acute myocardial infarction and acute ischemic heart disease that required hospitalization or revascularization by percutaneous coronary intervention or coronary artery bypass graft. Cerebrovascular diseases included transient cerebral ischemia and acute ischemic stroke that required hospitalization. The clinical conditions were ascertained by reviewing the International Classification of Diseases, Ninth Revision-Clinical Modification (ICD-9-CM) codes obtained from the National Health Insurance Research Database from 2001 to 2015, the latest information released with permission from the Health Promotion Administration at the Ministry of Health and Welfare in Taiwan (S3 Table). The cause of death was confirmed by reviewing Taiwan's National Death Registry.

## Covariates

Based on previous studies [18–20], we selected the following potentially significant covariates for the analysis: basic demographics (sex and age), personal behaviors (smoking status, alcohol consumption, and regular exercise habit), socioeconomic status (education level, average monthly income, and marital status), familial conditions (parental history of CVD), and clinical variable (LDL-C level). Data of all the covariates were collected from the structured questionnaire and biochemical tests administered at baseline. Details of all the covariates are presented in S4 Table.

## Statistical analyses

We combined the participants with obesity and overweight due to the small number of participants. Both metabolically healthy and unhealthy statuses were classified as normal weight and obesity/overweight. Thus, we had four groups: MHNW (the reference group), MHOO (metabolically healthy obesity/overweight), metabolically unhealthy normal weight, and metabolically unhealthy obesity/overweight. Descriptive analyses were performed on all data. Categorical and continuous variables were analyzed using the chi-square test and analysis of variance, respectively.

We calculated person-years for each participant at the beginning of the study and at occurrence of an event, death, or at December 31, 2015, whichever came first. The incidence rates of fatal and nonfatal CVD were calculated by dividing the number of cases by the number of 1,000-person-years of follow-up.

Kaplan–Meier survival curves were constructed to demonstrate the fatal and nonfatal CVD outcomes in the four groups over time. The log rank test was performed to compare the four groups. Cox regression analysis was performed to calculate multivariable adjusted hazard ratios (HRs) and 95% confidence intervals (CIs). The parallel lines of logarithm negative logarithm plot against logarithm of follow-up time indicate that the proportional hazards assumption was satisfied (S1 Fig) [21]. We applied following three models: baseline model 1, adjusted for sex and age; model 2, adjusted for socioeconomic factors such as smoking status, alcohol use, regular exercise, parental history of CVD, marital status, education level, and average monthly income; and model 3, adjusted for LDL-C level.

Potential effect modifiers—sex, age (cutoff, 65 years), and smoking status—were assessed by the likelihood ratio test to compare the goodness-of-fit of the models with and without the interaction terms in the fully adjusted model. Several sensitivity analyses were performed to test the robustness of our results. First, we excluded outcomes that occurred in the first year of follow-up to avoid possible inverse causal relationships. Second, body weight and height were self-reported in the first survey of TwSHHH and were measured in the second survey of TwSHHH. To calibrate under-reported BMI values from the survey, they had developed a formula. We redefined BMI using this formula from the TwSHHH database. Thirdly, we redefined the cardiometabolic diseases using a combination of prescriptions and the ICD-9-CM codes. Finally, we separately estimated the hazard ratio of the metabolically healthy obesity and metabolically healthy overweight.

All statistical tests were 2-tailed with a type I error of 0.05, and a *P* value of.05 was considered statistically significant. Analyses were performed using SAS software (version 9.4; SAS Institute, Cary, NC, USA) and Stata version 15 (Stata Corporation, College Station, TX, USA).

## Results

S2 Fig illustrates the flow diagram of the participant enrollment process. In this study, the follow-up rate was 99.4% of that in the National Health Interview Survey (NHIS) and missing data on covariates and laboratory tests were less than 0.5% and 15%, respectively. The baseline characteristics of 5 358 included individuals are listed in Table 1. The details information of participants with metabolically healthy obesity and overweight separately was in the S5 Table. The mean (standard deviation) age of the participants was 44.5 (15.3) years, and 48.2% of the participants were women. The prevalence of MHOO was 8.6% in the Taiwanese general population, which included individuals who were >20 years old, not pregnant, and did not have CVD (n = 5,719); however, the prevalence of MHOO was 22.2% in the population with obesity.

**Table 1. Baseline characteristics of participants by metabolic statuses and body mass index categories.**

| Characteristics | Metabolically healthy (n = 1,970) | | Metabolically unhealthy (n = 3,388) | | *P* value |
| --- | --- | --- | --- | --- | --- |
| | Normal weight | Obesity/overweight | Normal weight | Obesity/overweight | |
| | n = 1,479 | n = 491 | n = 1,668 | n = 1,720 | |
| | Mean (SD) | Mean (SD) | Mean (SD) | Mean (SD) | |
| Age (years old) | 38.1 (12.8) | 43.3 (13) | 46.2 (16.6) | 48.8 (14.5) | < .001 |
| Body mass index (kg/m$^2$) | 21.2 (1.5) | 26.4 (2.9) | 21.7 (1.5) | 27 (2.7) | < .001 |
| Waist circumference (cm) | 73.4 (6.7) | 85.9 (8.3) | 78.2 (7.7) | 90.1 (8.7) | < .001 |
| Systolic blood pressure (mmHg) | 105.6 (10) | 110.3 (9.6) | 120.4 (19.5) | 125.7 (17.7) | < .001 |
| Diastolic blood pressure (mmHg) | 69.2 (7.6) | 73.0 (6.8) | 77.3 (11.5) | 82 (11.4) | < .001 |
| Fasting plasma glucose (mg/dL) | 85.0 (6.7) | 87.6 (6.5) | 98.3 (33.7) | 104.2 (37.7) | < .001 |
| Hemoglobulin A1c (%) | 5.03 (0.47) | 5.17 (0.51) | 5.47 (1.28) | 5.74 (1.27) | < .001 |
| Total cholesterol (mg/dL) | 176.2 (28.1) | 183 (26.9) | 187.5 (41.8) | 198.1 (42.1) | < .001 |
| Triglycerides (mg/dL) | 83.3 (26.8) | 95.5 (27.8) | 142.1 (83.8) | 184.1 (105.7) | < .001 |
| High-density lipoprotein cholesterol (mg/dL) | 61.5 (11.6) | 60.8 (11.6) | 52.6 (16.3) | 49.6 (15.8) | < .001 |
| Low-density lipoprotein cholesterol (mg/dL) | 107.2 (20.6) | 113.7 (20.2) | 120.1 (29.2) | 127.3 (28.7) | < .001 |
| | n (%) | n (%) | n (%) | n (%) | |
| 20–39 (years old) | 862 (58.3) | 197 (40.1) | 638 (38.3) | 493 (28.7) | < .001 |
| 40–64 (years old) | 557 (37.7) | 264 (53.8) | 752 (45.1) | 945 (54.9) | < .001 |
| ≥65 (years old) | 60 (4.1) | 30 (6.1) | 278 (16.7) | 282 (16.4) | < .001 |
| Women | 850 (57.5) | 232 (47.3) | 822 (49.3) | 680 (39.5) | < .001 |
| Smokers | 297 (20.1) | 93 (18.9) | 433 (26) | 481 (28) | < .001 |
| Alcohol used | 358 (24.2) | 145 (29.5) | 475 (28.5) | 566 (32.9) | < .001 |
| Regular exercise habit[a] | 341 (23.1) | 131 (26.7) | 395 (23.7) | 421 (24.5) | .40 |
| Parental history of CVD | 287 (19.4) | 108 (22) | 367 (22) | 450 (26.2) | < .001 |
| Living with spouse | 880 (59.5) | 353 (71.9) | 1051 (63) | 1277 (74.2) | < .001 |
| Educational level (≥9 years of schooling) | 1019 (68.9) | 277 (56.4) | 909 (54.5) | 762 (44.3) | < .001 |
| Average monthly income ≥40,000 NTD | 307 (20.8) | 137 (27.9) | 343 (20.6) | 385 (22.4) | **.004** |

Body mass index categories: Normal weight, 18.5–23.9 kg/m$^2$; obesity/overweight, ≥24.0 kg/m$^2$.

[a]Regular exercise is defined as exercise for more than 30 minutes a day, three times a week, lasting at least three months.

Abbreviations: CVD, cardiovascular disease; NTD, New Taiwan Dollar; SD, standard deviation.

We documented 439 incident cases of fatal and nonfatal CVD in 67858.0 person-year during a median (interquartile range) follow-up time of 13.7 (13.6–13.8) years. The incident rate of fatal and nonfatal CVD was 6.47 per thousand person-year, including 259 carotid artery disease events, 209 stroke events, 4 fatal CAD events and 9 fatal strokes. A person may have had several events, but we used the first event to calculate the person year. The fatal and nonfatal CVD-free survival time was significantly different in the four groups (log rank test, *P* < .001). The Kaplan–Meier survival curves are presented in Fig 1. The adjusted fatal and nonfatal CVD HR with 95% CI was 1.74 (1.02, 2.99; Table 2).

Subgroup analysis (Table 3) revealed that fatal and nonfatal CVD risk was significantly higher in women with MHOO (adjusted HR: 4.65) than in men with MHOO (adjusted HR: 1.13). The fatal and nonfatal CVD risk in individuals with metabolically unhealthy profiles who were less than 65 years old was three to four folds, whereas in elderly individuals it was one to two folds. Smoking status did not change the point estimate significantly. In the subgroup analyses, in order to compare the fatal and nonfatal CVD risk between different sex and age groups, we changed the reference group to women with MHNW and, whom were less than 65 years old (S6 Table).

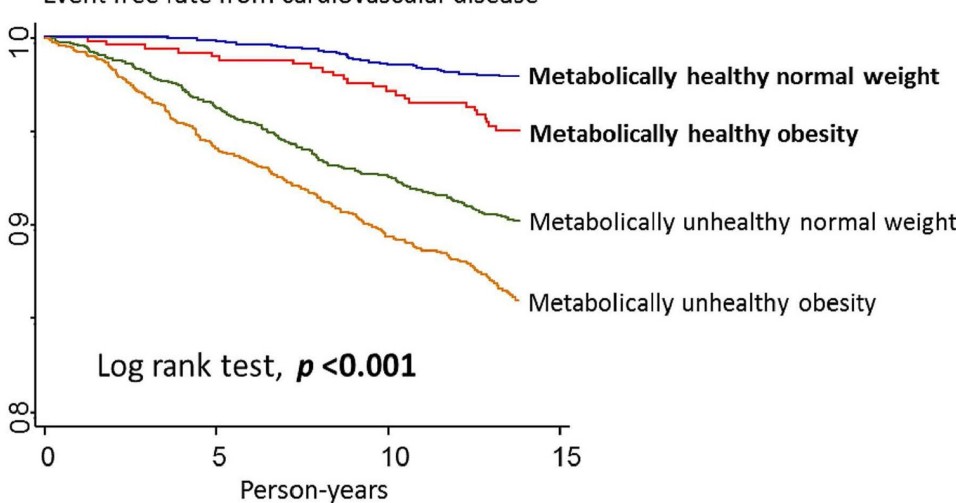

**Fig 1. Kaplan–Meier survival curves of fatal and nonfatal cardiovascular disease outcome by metabolic status and body mass index category.**

Sensitivity analysis results are presented in S7 and S8 Tables. In individuals with MHOO, the fatal and nonfatal CVD risk was found to be high after exclusion of events that occurred in the first year (adjusted HR: 1.74, 95% CI: 1.02, 2.99), redefining BMI using calibrated formulas (adjusted HR: 1.73, 95% CI: 1.01, 2.96), redefining cardiometabolic diseases (adjusted HR: 1.72, 95% CI: 1.01, 2.91), and redefining MHOO (adjusted HR: 1.70, 95% CI: 1.00, 2.88). However, the fatal and nonfatal CVD risk was not significantly in the metabolically healthy "obesity" group (adjusted HR: 0.97, 95% CI: 0.30, 3.19), only 3 events developed in the group, but remained significantly higher in the metabolically healthy "overweight" group (adjusted HR: 2.03, 95% CI: 1.17, 3.53) (S8 Table).

**Table 2. Risk of fatal and nonfatal cardiovascular disease by metabolic status and body mass index categories.**

| Variables | Metabolically healthy | | Metabolically unhealthy | |
|---|---|---|---|---|
| | Normal weight | Obesity/overweight | Normal weight | Obesity/overweight |
| Participants | 1,479 | 491 | 1,668 | 1,720 |
| Person-years | 19,786.7 | 6514.1 | 20,715.6 | 20,841.6 |
| Events | 30 | 24 | 156 | 229 |
| Incidence rate (per 1,000 person-years) | 1.52 | 3.70 | 7.53 | 10.97 |
| Model 1 | 1 | **1.74 (1.02, 2.99)** | **2.70 (1.82, 4.00)** | **3.50 (2.38, 5.14)** |
| Model 2 | 1 | **1.75 (1.02, 2.99)** | **2.64 (1.78, 3.92)** | **3.34 (2.27, 4.91)** |
| Model 3 | 1 | **1.74 (1.02, 2.99)** | **2.48 (1.65, 3.71)** | **3.09 (2.08, 4.58)** |

Body mass index categories: Normal weight, 18.5–23.9 kg/m$^2$; Obesity/Overweight, $\geq$24.0 kg/m$^2$.

Presented as hazard ratio and 95% confidence intervals.

Model 1: Adjusted for sex and age (20–39, 40–64, and $\geq$65 years old).

Model 2: Adjusted for smoking status (yes/no), alcohol use (yes/no) and regular exercise (yes/no), parental history of cardiovascular disease (yes/no), marital status (yes/no), educational level ($</\geq$ 9 years of schooling), and average monthly income ($</\geq$40,000 New Taiwan dollars).

Model 3: Adjusted for low-density lipoprotein cholesterol level.

Bold font indicates a significant risk.

**Table 3. Subgroup analyses for fatal and nonfatal cardiovascular disease risk by metabolic statuses and body mass index categories.**

| | Metabolically healthy | | Metabolically unhealthy | | |
|---|---|---|---|---|---|
| Variables | Normal weight | Obesity/overweight | Normal weight | Obesity/overweight | $P_{interaction}$ |
| **Sex** | | | | | **.010** |
| Women | 1 | **4.65 (1.61, 13.44)** | **5.77 (2.26, 14.70)** | **8.77 (3.49, 22.06)** | |
| Men | 1 | 1.13 (0.58, 2.22) | **1.83 (1.16, 2.88)** | **2.02 (1.29, 3.15)** | |
| **Age** | | | | | **.001** |
| <65 years old | 1 | 1.74 (0.83, 3.63) | **3.38 (1.94, 5.90)** | **4.24 (2.48, 7.26)** | |
| ≥65 years old | 1 | 1.89 (0.85, 4.18) | 1.50 (0.84, 2.69) | **1.88 (1.05, 3.35)** | |
| **Smoking status** | | | | | .65 |
| Smoker | 1 | 2.22 (0.80, 6.14) | **2.08 (1.03, 4.19)** | **2.40 (1.22, 4.70)** | |
| Non-smoker | 1 | 1.79 (0.94, 3.43) | **2.80 (1.69, 4.63)** | **3.54 (2.16, 5.79)** | |

Body mass index categories: Normal weight, 18.5–23.9 kg/m$^2$; obesity/overweight, ≥24.0 kg/m$^2$.

Model 3: Adjusted for low-density lipoprotein cholesterol level.

Presented as hazard ratio and 95% confidence interval.

Bold font indicates the significant risk factor.

## Discussion

Our data revealed that the prevalence of MHOO was 8.6% in the Taiwanese general population and 22.2% in the population with obesity, suggesting a positive association between MHOO and incidence of fatal and nonfatal CVD in Taiwan. This association was enhanced in women, suggesting that sex was a significant effect modifier. Our results were robust, suggesting that the relationship between MHOO and fatal and nonfatal CVD risk is convincing.

Accordingly, individuals with MHOO are not truly healthy if the definition is not sufficiently rigorous. We strictly defined MHOO by normal laboratory values and absence of hypertension, dyslipidemia, and diabetes; thus making our results more convincing. A previous study revealed that individuals with MHO in Taiwan had a significantly increased risk of hypertension, type 2 diabetes, and metabolic syndrome [13]. In addition, we confirm that individuals with MHOO have increased risk of fatal and nonfatal CVD outcome. Our results are compatible with those of previous meta-analyses demonstrating that individuals with MHO have a significantly higher risk of CVD; our point estimate was similar to their results: worldwide, 1.52 (HR) [6] and Asia, 1.61 (HR) [12].

Considering effect modifiers, previous studies have revealed that the prevalence of metabolic syndrome and CVD risk were higher in postmenopausal women than in men [22], which is consistent with our findings. Several articles with small sample size have not revealed sex differences [14, 23–25]. Our finding was supported by a study reporting that women with MHO have more hypertension [26]. Further MHOO studies focused on sex differences are warranted. Previous studies reported that younger individuals with MHO had a higher risk of CVD, which is consistent with our findings. A study indicated that the CVD risk can be as high as 5.1 folds in young men with MHO (mean age, 31.2 years) [27]. In addition, a study focused on older individuals (age, >55 years) with MHO indicated that the CVD risk was non-significant in all the participants [23]. Similarly, another study focused on older individuals (age, >60 years) reported that the CVD risk remained non-significant even after further stratification of participants by age (60–69, 70–79, and >80 years) [28]. A prevoius multivariate meta-regression analysis revealed that CVD risk in individuals with MHO significantly

decreased by 2% with increase in age each year [6]. The non-significant difference in our MHOO group may be explained by the small number of the elderly individuals.

Genetic or functional variants of adipocytes cause less chronic inflammation and changes in adipose tissue composition from subcutaneous to visceral fat with variation in gastrointestinal microbiota as well as early life programing and less sedentary lifestyle could explain the mechanism of MHOO [29, 30]. However, some people transition to a metabolically unhealthy state while others remain healthy. Adipocyte tissue dysfunction and cytokine secretion lead to systematic inflammation, insulin resistance, and ultimately, transformation into a metabolically unhealthy status with diabetes and atherogenic dyslipidemia. This transformation results in the development of CVD [31]. Nearly 33%–52% of individuals with MHO undergo transition to metabolically unhealthy status in 6–20 years [32]. Sex hormones, which act centrally as well as locally in adipocytes, exemplify significant differences in body weight, fat distribution, and energy balance between women and men. Sex as an effect modifier can be related to estrogen signaling through receptors, the G protein-coupled estrogen receptor 30 and other membrane-bound forms of the estrogen receptors, which serve as important mediators of cardiometabolic protective effects [33]. The attenuated CVD risk in elderly individuals with MHOO may be related to underlying comorbidities and sarcopenia. BMI is not a surrogate of excess fat, but of protective lean mass effect [34].

We can confidently claim that further health risks exist despite of the current MHOO status. We encourage every woman with overweight and obesity to maintain metabolically healthy and normal weight. In addition, we advocate adults to maintain metabolically healthy status in their early lifetime. Cardiometabolic risk and transition to MHOO status can be achieved by individuals with metabolically unhealthy obesity through 10% weight reduction, and further weight loss of 20% can result in transformation to the MHNW status and further reduce the CVD risk [7].

To the best of our knowledge, this study is the first to report fatal and nonfatal CVD risk in Taiwanese individuals with MHOO. This study has several strengths. First, enrollment of participants from a representative cohort study and long follow-up period reduced the possibility of selection bias and established the temporal relationship, respectively. Second, the exposure (MHOO status) and outcome (fatal and nonfatal CVD events) in our cohort study were ascertained by using a strict definition and by confirming the event occurrence from the National Health Insurance Research Database, respectively. Third, we confirmed the effect modifiers and performed several sensitivity analyses to confirm the robustness of our results. However, this study has some limitations. First, we combined the overweight and obesity groups, and thus, the dangerous group may not have been evaluated specifically. However, we had a large sample size to perform subgroup analyses and investigate effect modifiers. Second, we did not evaluate the changes in MHOO over time between the two TwSHHH populations and further transition or stability of MHOO is possible. Finally, our cohort was limited to the Taiwanese population and thus, external validity may be limited; however, we used a representative cohort to increase internal validity and avoid possible bias.

In conclusion, individuals with MHOO had significantly higher fatal and nonfatal CVD risk than those with MHNW. We provided evidence to encourage every individual with obesity to maintain a normal weight.

## Supporting information

**S1 Table. Definitions of hypertension, type 2 diabetes, hyperlipidemia in the study cohort.** (DOCX)

**S2 Table. The anatomical therapeutic chemical codes used to define the medications in the study cohort.**
(DOCX)

**S3 Table. Definitions of the clinical outcomes in the study cohort.**
(DOCX)

**S4 Table. Definitions of the covariates in the study cohort.**
(DOCX)

**S5 Table. Baseline characteristics of metabolically healthy obesity and metabolically healthy overweight participants.**
(DOCX)

**S6 Table. Subgroup analyses for fatal and nonfatal cardiovascular disease risk by metabolic statuses and body mass index categories by different reference groups.**
(DOCX)

**S7 Table. Sensitivity analyses of the risk of cardiovascular disease according to the metabolic status and anthropometric categories.**
(DOCX)

**S8 Table. Sensitivity analyses of the risk of fatal and nonfatal cardiovascular disease in metabolically healthy obesity and metabolically healthy overweight participants.**
(DOCX)

**S1 Fig. The logarithm negative logarithm plot against logarithm of time for proportional hazard assumption.**
(DOCX)

**S2 Fig. The flow diagram of the participants enrollment.**
(DOCX)

**S1 Raw data.**
(DOCX)

## Acknowledgments

We would like to thank Dr. You-Chen Lor (Hsinchu MacKay Memorial Hospital, Family Medicine department) for editing the manuscript; and Jing-Rong Jhuang (Institute of Epidemiology and Preventive Medicine, National Taiwan University) for statistical analysis. Further, we would like to thank to Pei-jin Li (librarian; MacKay Memorial Hospital) for examining the references.

## Author Contributions

**Conceptualization:** Tzu-Lin Yeh, Lee-Ching Hwang, Kuo-Liong Chien.

**Data curation:** Tzu-Lin Yeh, Hsin-Yin Hsu, Le-Yin Hsu.

**Formal analysis:** Tzu-Lin Yeh, Le-Yin Hsu.

**Methodology:** Tzu-Lin Yeh, Hsin-Yin Hsu, Ming-Chieh Tsai, Le-Yin Hsu.

**Software:** Tzu-Lin Yeh, Le-Yin Hsu.

**Supervision:** Kuo-Liong Chien.

**Validation:** Tzu-Lin Yeh, Hsin-Yin Hsu, Ming-Chieh Tsai.

**Writing – original draft:** Tzu-Lin Yeh.

**Writing – review & editing:** Lee-Ching Hwang, Kuo-Liong Chien.

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
