## [Decision Letter · Decision Letter 0]

3 Nov 2020

PONE-D-20-26474

Association between metabolically healthy obesity/overweight and cardiovascular disease risk : A representative cohort study in Taiwan

PLOS ONE

Dear Dr. Yeh,

Thank you for submitting your manuscript to PLOS ONE. After careful consideration, we feel that it has merit but does not fully meet PLOS ONE’s publication criteria as it currently stands. Therefore, we invite you to submit a revised version of the manuscript that addresses the points raised during the review process.

We look forward to receiving your revised manuscript.

Kind regards,

Martin Senechal, PhD

Academic Editor

PLOS ONE

Journal Requirements:

Reviewers' comments:

Reviewer's Responses to Questions

**Comments to the Author**

1. Is the manuscript technically sound, and do the data support the conclusions?

Reviewer #1: Yes

Reviewer #2: Yes

2. Has the statistical analysis been performed appropriately and rigorously? 

Reviewer #1: Yes

Reviewer #2: Yes

3. Have the authors made all data underlying the findings in their manuscript fully available?

Reviewer #1: No

Reviewer #2: Yes

4. Is the manuscript presented in an intelligible fashion and written in standard English?

Reviewer #1: Yes

Reviewer #2: Yes

5. Review Comments to the Author

Reviewer #1: This study examines risk of developing cardiovascular disease (CVD) in a cohort of people from Taiwan who were followed for ~14 years. The authors stratify subjects based on an index of metabolic health and whether they were normal weight or overweight/obese at baseline. It is reported that people that are metabolically healthy overweight/obese have a higher risk of developing CVD compared to people that were metabolically healthy normal weight at baseline. The magnitude of the increased CVD risk is in line with previous reports in other populations.

I have the following comments:

1) It is unclear why people with obesity and those that are overweight were combined. Was this due to low numbers of people with obesity in the study cohort?

2) MHO is used to mean both metabolically healthy obese and metabolically healthy overweight/obese. Perhaps the latter can be defined as MHOO to avoid confusion? Also obese is used is several places where people that are overweight are included.

3) In the Outcomes section study endpoints are CVD morbidity and mortality and cerebrovascular morbidity and mortality but I only see CVD morbidity data presented in the results section. If the Kaplan-Meier curves show mortality data this should be clearer as it reads as if these report time without CVD.

4) The authors highlight differences in responses between women and men and in young vs older subjects. However, it is likely that the baseline risk was higher in men vs women and in older vs younger adults, which is lost when 1 is used as the reference value for all. It would be helpful to recalculate these risks all compared to the group with the lowest risk. So in Table 3 for the sex analysis all groups would likely be compared to women MHNW and for Age compared to the <65 year old MHNW group. Presenting results from this additional analysis would help the interpretation of the data.

5) In the discussion what support is there in humans that adipose tissue inflammation causes systemic inflammation and what “changes in adipose tissue composition” are there? Please be specific.

6) In the discussion the statement “the protective status of MHO is temporary” is too strong. It is true, and the authors mention later in the paragraph, that some people with MHO transition to metabolically unhealthy obesity over time but not all do.

7) Why was LDL-cholesterol included as a covariate when this was included in the metabolically healthy criteria? Please include the rationale.

8) Why was 2015 used as the study end point?

9) Abstract – clarify that 8.6% is the prevalence of metabolically healthy overweight/obese in all subjects.

10) Waist circumference – In the abstract waist circumference is mentioned as being part of the inclusion criteria while in the methods section WC is mentioned as not being included. Which is correct?

11) Why was a cut-off of 160 mg/dl used for LDL-cholesterol? Less than 130 mg/dl is typically used as a normal concentration.

12) Does “current smoking status” only relate to when the study started in 2002? Use of “current” make this confusing.

Reviewer #2: This is a well-structured study with a rigorous analysis. I have only several minor comments:

(1) In line 100, the authors stated that "based on previous studies, ...", the references were lacking, please provide.

(2) In Results sections, the description was too simplified, such as lines 161-164. I suggest to develop more in detail.

(3) Even though the authors indicated the limitation of not dividing obesity and overweight participants, I strongly comment to conduct a sensitivity analysis to evaluate the standard MHO groups without combining the overweight people.

6. PLOS authors have the option to publish the peer review history of their article (what does this mean?). If published, this will include your full peer review and any attached files.

Reviewer #1: **Yes: **Gordon Smith

Reviewer #2: No

---

## [Author Response · Author response to Decision Letter 0]

9 Dec 2020

PONE-D-20-26474

Association between metabolically healthy obesity/overweight and cardiovascular disease risk : A representative cohort study in Taiwan

PLOS ONE

Journal Requirements:

Response：Thanks for your advice. We revised our manuscript and references to meet the PLOS ONE's style requirements.

 Response：Thanks for your advice. We did not receive any funding for this study and stated “The authors received no specific funding for this work” in the cover letter.

 Response：Thanks for your advice. The datasets generated and/or analysed during the current study are not publicly available due to the terms of consent to which the participants agreed but data are however available from the authors upon reasonable request and with permission of the Health Promotion Administration at the Ministry of Health and Welfare in Taiwan. We revised our cover letter and provided contact information for a data access committee, ethics committee, or other institutional body to which data requests may be sent.

Reviewers' comments: 

1. Is the manuscript technically sound, and do the data support the conclusions?

Reviewer #1: Yes Reviewer #2: Yes

2. Has the statistical analysis been performed appropriately and rigorously?

Reviewer #1: Yes Reviewer #2: Yes

3. Have the authors made all data underlying the findings in their manuscript fully available?

Reviewer #1: No Reviewer #2: Yes

4. Is the manuscript presented in an intelligible fashion and written in standard English?

Reviewer #1: Yes Reviewer #2: Yes

5. Review Comments to the Author

Reviewer #1: This study examines risk of developing cardiovascular disease (CVD) in a cohort of people from Taiwan who were followed for ~14 years. The authors stratify subjects based on an index of metabolic health and whether they were normal weight or overweight/obese at baseline. It is reported that people that are metabolically healthy overweight/obese have a higher risk of developing CVD compared to people that were metabolically healthy normal weight at baseline. The magnitude of the increased CVD risk is in line with previous reports in other populations.

I have the following comments:

1) It is unclear why people with obesity and those that are overweight were combined. Was this due to low numbers of people with obesity in the study cohort?

Response：Thanks for your expert opinion. We combined people with obesity and those that are overweight due to low numbers of people with obesity in the study cohort. We revised our manuscript and added the baseline characteristics of metabolically healthy obesity and overweight separately in Table S5 and performed a sensitivity analysis according to the reviewer 2’s suggestion in Table S6.

Revised methods: Both metabolically healthy and unhealthy statuses were classified as normal weight and obesity/overweight. We combined the participants with obesity and overweight due to the small number of participants….

The baseline characteristics of 5 358 included individuals are listed in Table 1. The details information of obesity and overweight separately was in the S5 Table. 

Table S5. Baseline characteristics of metabolically healthy obesity and overweight participants 

 Metabolically healthy (n =491) Metabolically healthy 

(n =1,720) 

Characteristics Overweight

(n= 358) Obesity

(n= 133) Overweight

(n= 1,054) Obesity

(n= 666) p value

 Mean (SD) Mean (SD) Mean (SD) Mean (SD) 

Age (years old) 44.6 (13.1) 40.1 (12.4) 49.6 (14.1) 47.5 (15.1) <0.001

Body mass index (kg/m2 ) 25.2 (0.8) 29.6 (4.0) 25.4 (0.9) 29.7 (2.5) <0.001

Waist circumference (cm) 83.5 (6.7) 92.5 (8.4) 86.6 (6.9) 95.7 (8.4) <0.001

Systolic blood pressure (mmHg) 109.4 (9.7) 112.9 (8.9) 124.1 (17.6) 128.2 (17.6) <0.001

Diastolic blood pressure (mmHg) 72.5 (6.9) 74.2 (6.4) 80.7 (11.2) 84.0 (11.3) <0.001

Fasting plasma glucose (mg/dL) 87.4 (6.4) 88.1 (6.6) 102.2 (35.1) 107.4 (41.3) <0.001

Hemoglobulin A1c (%) 5.15 (0.52) 5.22 (0.48) 5.66 (1.24) 5.86 (1.32) <0.001

Total cholesterol (mg/dL) 182.2 (26.2) 184.8 (28.6) 196.3 (40.6) 200.8 (44.3) <0.001

Triglycerides (mg/dL) 94.1 (28.1) 99.5 (26.9) 178.0 (102.7) 193.8 (109.8) <0.001

High-density lipoprotein cholesterol (mg/dL) 60.8 (11.5) 60.8 (11.8) 50.1 (15.7) 48.8 (15.8) <0.001

Low-density lipoprotein cholesterol (mg/dL) 113.2 (20.0) 114.9 (20.5) 126 (28.4) 129.3 (29.1) <0.001

 n (%) n (%) n (%) n (%) 

20-39 (years old) 134 (37.4) 63 (47.0) 273 (25.9) 220 (33.1) <0.001

40-64 (years old) 197 (54.9) 68 (50.8) 599 (56.9) 345 (51.9) <0.001

≥ 65 (years old) 28 (7.8) 3 (2.3) 181 (17.2) 100 (15.0) <0.001

Women 194 (54.0) 67 (50.0) 637 (60.4) 403 (60.5) <0.001

Current smokers 63 (17.6) 31 (23.1) 284 (27.0) 196 (29.5) <0.001

Alcohol used 110 (30.6) 35 (26.1) 348 (33.1) 218 (32.8) <0.001

Regular exercise habita 101 (28.2) 30 (22.4) 277 (26.3) 144 (21.7) <0.001

Menopaused status 56 (15.6) 21 (15.7) 227 (21.6) 149 (22.4) 0.008

Parental history of CVD 87 (24.2) 21 (15.8) 291 (27.6) 159 (23.9) <0.001

Living with spouse 269 (74.9) 86 (64.2) 804 (76.4) 471 (70.8) <0.001

Educational level ( ≥ 9 years of schooling) 203 (56.6) 75 (56.0) 466 (44.3) 295 (44.4) <0.001

Average month income ≥ 40,000 NTD 110 (30.6) 28 (20.9) 246 (23.4) 138 (20.8) <0.001

Normal weight, 18.5 to 23.9 kg/m2; obesity/overweight, ≥24.0 kg/m2

aRegular exercise defined as more than 30 minutes a day, three times a week, lasting at least three months

SD: standard deviation 

Table S6. Subgroup analyses for fatal and nonfatal cardiovascular disease risk by metabolic statuses and body mass index categories by different reference groups.

 Metabolically healthy Metabolically unhealthy

Variables Normal weight Obesity/overweight Normal weight Obesity/overweight

Sex 

 Women 1 5.00 (1.74, 14.41) 5.77 (2.28, 14.59) 8.82 (3.54, 21.92)

 Men 7.42 (2.81, 19.58) 8.96 (3.21, 25.07) 13.78 (5.51, 34.45) 15.15 (6.11, 37.54)

Age 

 <65 years old 1 2.06 (0.99, 4.28) 4.47 (2.57, 7.78) 5.65 (3.31, 9.63)

 ≥65 years old 16.04 (7.75, 33.22) 30.74 (14.42, 65.57) 21.63 (12.32, 37.97) 24.94 (14.30, 43.50)

2) MHO is used to mean both metabolically healthy obese and metabolically healthy overweight/obese. Perhaps the latter can be defined as MHOO to avoid confusion? Also obese is used is several places where people that are overweight are included.

Response：Thanks for your expert opinion. We revised our manuscript and changed the word MHO to MHOO to avoid confusion.

Revised sentence: Thus, we had four groups: MHNW (the reference group), MHOO (metabolically healthy obesity/overweight), …

3) In the Outcomes section study endpoints are CVD morbidity and mortality and cerebrovascular morbidity and mortality but I only see CVD morbidity data presented in the results section. If the Kaplan-Meier curves show mortality data this should be clearer as it reads as if these report time without CVD.

Response：Thanks for your expert opinion. We revised our manuscript and changed the CVD morbidity data presented in the results section to the CVD morbidity and mortality. 

Revised sentence: We documented 439 incident cases of fatal and nonfatal CVD during a median… . The fatal and nonfatal CVD-free survival time … The adjusted fatal and nonfatal CVD HR ….

We revised the title of the figures and tables:

Fig 1. Kaplan–Meier survival curves of fatal and nonfatal cardiovascular disease outcome by metabolic status and body mass index category.

Table 2. Risk of fatal and nonfatal cardiovascular disease by metabolic status and body mass index categories.

Table 3. Subgroup analyses for fatal and nonfatal cardiovascular disease risk by metabolic statuses and body mass index categories.

4) The authors highlight differences in responses between women and men and in young vs older subjects. However, it is likely that the baseline risk was higher in men vs women and in older vs younger adults, which is lost when 1 is used as the reference value for all. It would be helpful to recalculate these risks all compared to the group with the lowest risk. So in Table 3 for the sex analysis all groups would likely be compared to women MHNW and for Age compared to the <65 year old MHNW group. Presenting results from this additional analysis would help the interpretation of the data.

Response：Thanks for your expert opinion. We revised our manuscript and changed the reference group in S7 Table.

Results: The fatal and nonfatal CVD risk in individuals with metabolically unhealthy profiles who were less than 65 years old was three to four folds, whereas in elderly individuals it was one to two folds. Smoking status did not change the point estimate significantly. In the subgroup analyses, in order to compare the fatal and nonfatal CVD risk between different sex and age groups, we changed the reference group to women with MHNW and, whom were less than 65 years old (S6 Table).

Table S6. Subgroup analyses for fatal and nonfatal cardiovascular disease risk by metabolic statuses and body mass index categories by different reference groups.

 Metabolically healthy Metabolically unhealthy

Variables Normal weight Obesity/overweight Normal weight Obesity/overweight

Sex 

 Women 1 5.00 (1.74, 14.41) 5.77 (2.28, 14.59) 8.82 (3.54, 21.92)

 Men 7.42 (2.81, 19.58) 8.96 (3.21, 25.07) 13.78 (5.51, 34.45) 15.15 (6.11, 37.54)

Age 

 <65 years old 1 2.06 (0.99, 4.28) 4.47 (2.57, 7.78) 5.65 (3.31, 9.63)

 ≥65 years old 16.04 (7.75, 33.22) 30.74 (14.42, 65.57) 21.63 (12.32, 37.97) 24.94 (14.30, 43.50)

5) In the discussion what support is there in humans that adipose tissue inflammation causes systemic inflammation and what “changes in adipose tissue composition” are there? Please be specific.

Response：Thanks for your expert opinion. We revised our discussion: 

Genetic or functional variants of adipocytes cause less chronic inflammation and changes in adipose tissue composition from subcutaneous to visceral fat with variation in gastrointestinal microbiota as well as early life programing and less sedentary lifestyle could explain the mechanism of MHOO.[26, 27]

6) In the discussion the statement “the protective status of MHO is temporary” is too strong. It is true, and the authors mention later in the paragraph, that some people with MHO transition to metabolically unhealthy obesity over time but not all do.

Response：Thanks for your expert opinion. We revised our discussion: …However, the protective unharmful status of MHOO is temporary….

7) Why was LDL-cholesterol included as a covariate when this was included in the metabolically healthy criteria? Please include the rationale.

Response：Thanks for your expert opinion. In the metabolically healthy criteria, we exclude participants with existed hyperlipidemia, referred to a LDL-C level of ≥160 mg/dL in the first survey or long-term prescription of lipid-lowering agents. In the model 3, LDL-C level was served as a continuous variable, to adjust the different levels below 160 mg/dL. We added the description in the Table S4. 

Table S4. Definitions of the covariates in the study cohort

Covariates

Question in the questionnaire Category

Sex Women/Men

Age (Years old) ≥20 to <40, ≥40 to <65, ≥65

… … …

Low-density lipid cholesterol (mg/dL) Continuous variable

8) Why was 2015 used as the study end point?

Response：Thanks for your expert opinion. This is the limitation of our cohort data, we revised the method: The clinical conditions were ascertained by reviewing the International Classification of Diseases, Ninth Revision-Clinical Modification (ICD-9-CM) codes obtained from the National Health Insurance Research Database from 2001 to 2015, the latest information released with permission from the Health Promotion Administration at the Ministry of Health and Welfare in Taiwan. (S3 Table).

9) Abstract – clarify that 8.6% is the prevalence of metabolically healthy overweight/obese in all subjects.

Response：Thanks for your expert opinion. We revised our abstract:

Among 5 358 subjects (mean [standard deviation] age, 44.5 [15.3] years; women, 48.2%), 1 479 were metabolically healthy with normal weight and 491 were metabolically healthy with obesity, with a prevalence of 8.6%. The prevalence of MHOO was 8.6% in the Taiwanese general population, which included individuals who were >20 years old, not pregnant, and did not have CVD (n = 5,719).

10) Waist circumference – In the abstract waist circumference is mentioned as being part of the inclusion criteria while in the methods section WC is mentioned as not being included. Which is correct?

Response：Thanks for your expert opinion. We revised our abstract:

Participants without diabetes, hypertension, and hyperlipidemia and who did not meet the metabolic syndrome without and waist circumference criteria were considered metabolically healthy.

11) Why was a cut-off of 160 mg/dl used for LDL-cholesterol? Less than 130 mg/dl is typically used as a normal concentration.

Response：Thanks for your expert opinion. According to the ATP III guidelines, LDL-c higher than 160 mg/dl was high and was more likely to be concerned as a disease status; LDL-c in130-159 mg/dl was considered as borderline high. We intended to exclude the hyperlipidemia disease status for the general population, thus we chose the cut-off of 160 mg/dl rather than 130 mg/dl. 

12) Does “current smoking status” only relate to when the study started in 2002? Use of “current” make this confusing.

Response：Thanks for your expert opinion. The “current smoking status” only relate to when the study started in 2002, and we deleted the “current” in our manuscript.

 Table S4. Definitions of the covariates in the study cohort

Covariates

Question in the questionnaire Category

Sex Women/Men

…. … …

Current Smoking status Have you smoked before? 1. Never; 2. Yes, but only a few times;…. 1, 2 =current smoker Current/Non-current smoker

Smoker/Non-smoker

Table 3. Subgroup analyses for fatal and nonfatal cardiovascular disease risk by metabolic statuses and body mass index categories.

 Metabolically healthy Metabolically unhealthy

Variables Normal weight Obesity/overweight Normal weight Obesity/overweight

 Smoker 1 2.22 (0.80, 6.14) 2.08 (1.03, 4.19) 2.40 (1.22, 4.70)

 Non-smoker 1 1.79 (0.94, 3.43) 2.80 (1.69, 4.63) 3.54 (2.16, 5.79)

Reviewer #2: This is a well-structured study with a rigorous analysis. I have only several minor comments:

(1) In line 100, the authors stated that "based on previous studies, ...", the references were lacking, please provide.

Response：Thanks for your expert opinion. We revised our method: 

Based on previous studies[18-20], we selected the following potentially significant covariates for the analysis: basic demographics (sex and age).

Reference:

18. Thomsen M, Nordestgaard BG. Myocardial infarction and ischemic heart disease in overweight and obesity with and without metabolic Syndrome. JAMA Internal Medicine. 2014;174(1):15-22. doi: 10.1001/jamainternmed.2013.10522.

19. Ortega FB, Lee DC, Katzmarzyk PT, Ruiz JR, Sui X, Church TS, et al. The intriguing metabolically healthy but obese phenotype: Cardiovascular prognosis and role of fitness. European Heart Journal. 2013;34(5):389-97. doi: 10.1093/eurheartj/ehs174.

20. Appleton SL, Seaborn CJ, Visvanathan R, Hill CL, Gill TK, Taylor AW, et al. Diabetes and cardiovascular disease outcomes in the metabolically healthy obese phenotype: A cohort study. Diabetes Care. 2013;36(8):2388-94. doi: 10.2337/dc12-1971.

(2) In Results sections, the description was too simplified, such as lines 161-164. I suggest to develop more in detail.

Response：Thanks for your expert opinion. We revised our results: 

We documented 439 incident cases of fatal and nonfatal CVD in 67858.0 person-year during a median (interquartile range) follow-up time of 13.7 (13.6–13.8) years. The incident rate of fatal and nonfatal CVD was 6.47 per thousand person-year, including 259 carotid artery disease events, 209 stroke events, 4 fatal CAD, 9 fatal stroke, a person may had several events, but we calculated the first event as the person year. The fatal and nonfatal CVD-free survival time was significantly different in the four groups…

(3) Even though the authors indicated the limitation of not dividing obesity and overweight participants, I strongly comment to conduct a sensitivity analysis to evaluate the standard MHO groups without combining the overweight people.

Response：Thanks for your expert opinion. We conducted a sensitivity analysis to evaluate the standard MHO groups without combining the overweight people, only 3 events in the standard MHO group was noted.

Revised method: Several sensitivity analyses were performed to test the robustness of our results.… Finally, we separately estimated the hazard ratio of the metabolically healthy obesity and metabolically healthy overweight.

Revised result: Sensitivity analysis results are presented in Table S7-8 Tables..... However, the fatal and nonfatal CVD risk was not significantly in the metabolically healthy “obesity ” group (adjusted HR: 0.97, 95% CI: 0.30, 3.19), only 3 events developed in the group, but remained significantly higher in the metabolically healthy “overweight” group (adjusted HR: 2.03, 95% CI: 1.17, 3.53) (S8 Table).

Table S8. Sensitivity analyses of the risk of fatal and nonfatal cardiovascular disease in metabolically healthy obesity and metabolically healthy overweight participants.

　 Metabolically healthy Metabolically unhealthy

Variables Normal weight Overweight Obesity Normal weight Obesity/overweight

Participants 1,479 358 133 1,668 1,720

Person-years 19,786.7 4722.4 1799.6 20,715.6 20,841.6

Events 30 23 3 156 229

Incidence rate

 (per 1,000 person-years) 1.52 4.87 1.67 7.53 10.97

Model 1 1 2.00 (1.16, 3.48) 1.02 (0.31, 3.35) 2.70 (1.82, 4.00) 3.50 (2.38, 5.14)

Model 2 1 2.02 (1.16, 3.51) 0.99 (0.30, 3.26) 2.64 (1.78, 3.92) 3.34 (2.27, 4.91)

Model 3 1 2.03 (1.17, 3.53) 0.97 (0.30, 3.19) 2.48 (1.65, 3.71) 3.09 (2.08, 4.58)

---

## [Decision Letter · Decision Letter 1]

31 Dec 2020

PONE-D-20-26474R1

Association between metabolically healthy obesity/overweight and cardiovascular disease risk : A representative cohort study in Taiwan

PLOS ONE

Dear Dr. Yeh,

Thank you for submitting your manuscript to PLOS ONE. After careful consideration, we feel that it has merit but does not fully meet PLOS ONE’s publication criteria as it currently stands. Therefore, we invite you to submit a revised version of the manuscript that addresses the points raised during the review process.

Please address reviewer #1 comments and pay attention to the MHOO acronyms in the manuscript. Make sure to spell it out at first appearance and then use the acronyms throughout.

We look forward to receiving your revised manuscript.

Kind regards,

Martin Senechal, PhD

Academic Editor

PLOS ONE

Reviewers' comments:

Reviewer's Responses to Questions

**Comments to the Author**

1. If the authors have adequately addressed your comments raised in a previous round of review and you feel that this manuscript is now acceptable for publication, you may indicate that here to bypass the “Comments to the Author” section, enter your conflict of interest statement in the “Confidential to Editor” section, and submit your "Accept" recommendation.

Reviewer #1: (No Response)

2. Is the manuscript technically sound, and do the data support the conclusions?

Reviewer #1: Yes

3. Has the statistical analysis been performed appropriately and rigorously? 

Reviewer #1: Yes

4. Have the authors made all data underlying the findings in their manuscript fully available?

Reviewer #1: Yes

5. Is the manuscript presented in an intelligible fashion and written in standard English?

Reviewer #1: Yes

6. Review Comments to the Author

Reviewer #1: The manuscript has been improved and the authors have adequately addressed my concerns with the exception of one minor concern listed below:

1) In response to the following comment “In the discussion the statement “the protective status of MHO is temporary” is too strong. It is true, and the authors mention later in the paragraph, that some people with MHO transition to metabolically unhealthy obesity over time but not all do.” The authors changed the sentence to read “However, the unharmful status of MHOO is temporary.” This still doesn’t reflect the fact that some MHOOs remain healthy over time and it isn’t “temporary”. Adding information that some people transition to a metabolically unhealthy state while others remain healthy would help clarify this statement.

7. PLOS authors have the option to publish the peer review history of their article (what does this mean?). If published, this will include your full peer review and any attached files.

Reviewer #1: No

---

## [Author Response · Author response to Decision Letter 1]

3 Jan 2021

6. Review Comments to the Author

Reviewer #1: The manuscript has been improved and the authors have adequately addressed my concerns with the exception of one minor concern listed below:

1) In response to the following comment “In the discussion the statement “the protective status of MHO is temporary” is too strong. It is true, and the authors mention later in the paragraph, that some people with MHO transition to metabolically unhealthy obesity over time but not all do.” The authors changed the sentence to read “However, the unharmful status of MHOO is temporary.” This still doesn’t reflect the fact that some MHOOs remain healthy over time and it isn’t “temporary”. Adding information that some people transition to a metabolically unhealthy state while others remain healthy would help clarify this statement.

Response：Thanks for your expert opinion. We revised our manuscript and changed the paragraph:

However, the unharmful status of MHOO is temporary some people transition to a metabolically unhealthy state while others remain healthy. Adipocyte tissue dysfunction and cytokine secretion lead to systematic inflammation, insulin resistance, and ultimately, transformation into a metabolically unhealthy status with diabetes and atherogenic dyslipidemia. This transformation results in the development of CVD. [31] Nearly 33%–52% of individuals with MHO undergo transition to metabolically unhealthy status in 6–20 years. [32]

---

## [Editor Report · Decision Letter 2]

18 Jan 2021

Association between metabolically healthy obesity/overweight and cardiovascular disease risk : A representative cohort study in Taiwan

PONE-D-20-26474R2

Dear Dr. Yeh,

We’re pleased to inform you that your manuscript has been judged scientifically suitable for publication and will be formally accepted for publication once it meets all outstanding technical requirements.

Kind regards,

Martin Senechal, PhD

Academic Editor

PLOS ONE
---

## [Editor Report · Acceptance letter]

21 Jan 2021

PONE-D-20-26474R2 

Association between metabolically healthy obesity/overweight and cardiovascular disease risk: A representative cohort study in Taiwan 

Dear Dr. Yeh:

I'm pleased to inform you that your manuscript has been deemed suitable for publication in PLOS ONE. Congratulations! Your manuscript is now with our production department. 

Kind regards, 

on behalf of

Dr. Martin Senechal 

Academic Editor

PLOS ONE